# TEST-TIME SCALING WITH REFLECTIVE GENERATIVE MODEL

**Zixiao Wang**[1*]  **Yuxin Wang**[2*]  **Xiaorui Wang**[2]  **Mengting Xing**[2]
**Jie Gao**[2]  **Jianjun Xu**[1]  **Guangcan Liu**[2]  **Chenhui Jin**[1]
**Zhuo Wang**[1]  **Shengzhuo Zhang**[2]  **Hongtao Xie**[1†]

[1]University of Science and Technology of China    [2]MetaStone Technology

wzx99@mail.ustc.edu.cn, wangyx58@ustc.edu.cn, htxie@ustc.edu.cn

## ABSTRACT

We introduce a new Reflective Generative Model (RGM), which obtains OpenAI o3-mini's performance via a novel Reflective Generative Form. This form focuses on high-quality reasoning trajectory selection and contains two novelties: 1) **A unified interface for policy and process reward model**: we share the backbone network and use task-specific heads for reasoning trajectory predicting and scoring respectively, introducing only 50M extra parameters for trajectory scoring. 2) **Eliminating the reliance on process-level annotation**: we provide a self-supervised process reward model (SPRM), which can directly learn the high-quality reasoning trajectory selection from the outcome reward. Equipped with the reflective generative form, RGM is naturally suitable for test-time scaling based on the controllable thinking length. Experiments show that our RGM, equipped with only 50M additional parameters in SPRM, outperforms policy models with 72B extra reward models, thereby enabling 32B model to outperform OpenAI o3-mini on AIME24 (84.2 vs. 79.6) and HMMT25 (53.1 vs. 53.0). Code is available at https://github.com/MetaStone-AI/XBai-o4.

## 1 INTRODUCTION

Over the past two years, the field of Large Language Models (LLMs) has experienced rapid advancements, marked by the emergence of increasingly sophisticated models. Notable developments include OpenAI's GPT-4, Google's Gemini, Meta's LLaMA series, Alibaba's Qwen, and DeepSeek's R1, which have collectively pushed the boundaries of natural language understanding and generation. This progress is attributed to innovations in model architectures and training techniques, enabling LLMs to process and generate content across various formats.

Recent analyses suggest that OpenAI's o3 model achieves its advanced reasoning and coding capabilities through Test-Time Scaling (TTS) techniques such as massive sampling, candidate scoring, and search over multiple reasoning paths (Labs, 2025; Zeff, 2024). For instance, during ARC-AGI and competitive coding evaluations, o3 was shown to generate up to 1024 candidate samples for each query (Chollet, 2024; OpenAI, 2025). These inference-time strategies mark a significant shift from traditional one-pass models, enabling o3 to adapt dynamically to novel tasks and achieve near-human performance in reasoning benchmarks.

TTS approaches can be categorized into two types: internal TTS and external TTS. Internal TTS (also called sequential TTS in Zeng et al. (2025)) strategies use CoT for longer thinking processes (Guo et al., 2025; OpenAI, 2024), which benefits from Long-CoT Supervised Fine-Tuning and reinforcement learning. Recent internal TTS methods (Guo et al., 2025) mainly suffer from the false positive reasoning process, as the outcome reward will misclassify the correct answer with incorrect reasoning during the training stage. External TTS (also called parallel TTS in Zeng et al. (2025)) is proposed for selecting the correct reasoning process. Prominent external TTS algorithms include

---

*Equal contribution.
†Corresponding author.

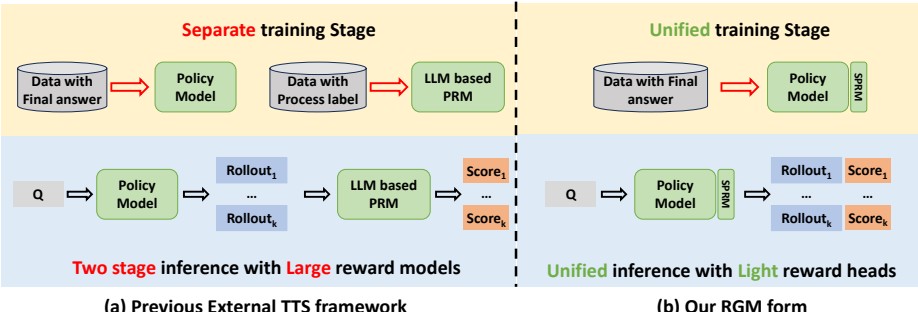

Figure 1: Comparison between the previous external TTS framework (a) and our RGM (b).

Best-of-N sampling, Beam Search, and Diverse Verifier Tree Search, using the reward model as the verifier to select high-quality reasoning trajectories. Researchers (Lightman et al., 2023) have shown that the Process Reward Model (PRM) is more effective in performance boosting compared with the Outcome Reward Model (ORM). However, Wang et al. (2023); Guan et al. (2025) point out that training a high-quality PRM remains costly, primarily due to the lack of accurate process-level annotations. Moreover, during the inference stage, introducing an additional LLM-based PRM introduces significant extra parameters and computational overhead, which severely limits the practical deployment of external TTS.

This paper focuses on external TTS and proposes a new Reflective Generative Form for high-quality reasoning trajectory selection. Specially, the proposed new form shares the backbone of the policy model and process reward model, providing a more efficient scoring process with little parameter and computational overhead. Besides, a Self-supervised Process Reward Mode(SPRM) is introduced for self-supervised training to eliminate the reliance on process-level supervision. Based on the Reflective Generative Form, the proposed RGM can improve the performance by increasing the controllable thinking length during inference. Compared with the existing external TTS framework with LLM based PRM, our proposed RGM introduces a unified form for both training and inference, achieving a more streamlined pipeline with significantly reduced computational and parameter overhead (Fig. 1). Experiment results show that RGM achieves comparable performance to OpenAI o1-mini and o3-mini with 7B and 32B parameters. And our SPRM with only 50M parameters outperforms existing 72B level reward models.

In summary, the main contributions of this paper are as follows:

- We provide **a new Reflective Generative Form** for high-quality reasoning trajectory selection, which enables a single network to achieve both reasoning trajectory prediction and selection with **Zero** process-level annotation.

- We provide both qualitative and quantitative analysis for **the aha moment** and **generalization capability** of the proposed new form. These exhaustive discussions will effectively benefit the community for future research.

- RGM outperforms existing 72B reward models with only **50M SPRM**, and achieves comparable performance to OpenAI o3-mini with only 32B policy models.

## 2 RELATED WORKS

**Test-Time Scaling.** Test-Time Scaling (TTS) is a technique that leverages additional computational resources at inference time to tackle challenging problems. TTS can be divided into two categories: internal TTS and external TTS. Internal TTS introduces the long Chain-of-Thought (CoT) to generate answers based on the detailed reasoning process. OpenAI o1(Jaech et al., 2024) and DeepSeek R1(Guo et al., 2025) introduce a thinking process to plan the solution and guide the final answer. Jin et al. (2024); Yeo et al. (2025) have shown that long CoT can help models correct mistakes by themselves and decompose complex problems more effectively. However, Chen et al. (2024b;a) have highlighted the risk of overthinking, where excessively long reasoning trajectories may lead to performance degradation. On the other hand, external TTS scales up inference through

search-based strategies and auxiliary reward models. A common approach is the Best-of-N strategy (Lightman et al., 2023; Brown et al., 2024; Wang et al., 2023). Fine-grained step level searching methods have also been explored, such as Beam Search (Liu et al., 2025; Snell et al., 2024), Diverse Verifier Tree Search (Beeching et al.) and Monte Carlo Tree Search (MCTS) (Zhang et al., 2024; Guan et al., 2025; Luo et al., 2024). These methods search at the step level and utilize Process Reward Models (PRMs) to guide the reasoning trajectory step-by-step. Beyond search strategies, recent work emphasizes that the quality of the reward model is a crucial factor in external TTS (Guan et al., 2025).

**Process Reward Model.**  Process Reward Models (PRMs) focus on evaluating LLMs at the step level. Lightman et al. (2023) unveil that this fine-grained guidance can lead to better TTS performance compared with the global-level Outcome Reward Model (ORM). However, accurately identifying logical errors in LLM outputs remains challenging, and PRMs require high-quality task-specific annotated data for training. To this end, recent works Wang et al. (2023) leverage Monte Carlo estimation to automatically assign step-level scores using only the final answers as supervision. Zhang et al. (2024); Guan et al. (2025) iteratively synthesizes data by MCTS and fine-tuning both LLMs and PRMs, improving performance across both models. Tan et al. (2025) follow the LLM-as-a-judge method and introduce a new LLM to annotate the reward of each step. Nonetheless, Zhang et al. (2025) point out that labels generated by Monte Carlo estimation can be noisy, as incorrect reasoning processes may still yield correct final answers. They further propose a hybrid approach that combines both Monte Carlo estimation with the LLM-as-a-judge.

## 3  PROBLEM FORMULATION

This paper aims to find a high-quality reasoning trajectory more efficiently at inference time based on TTS. We first summarize the general inference forms for standard LLMs (policy models) and existing TTS methods, and then formally define our proposed Reflective Generative Form.

**1) LLMs without TTS.** The model directly generates an answer based on the input query Q. This basic inference form can be formulated as:

$$\text{answer} = LLM_{\text{answer}}(\mathbf{Q}). \tag{1}$$

TTS based methods can be categorized into two types: sequential scaling based internal TTS and parallel scaling based external TTS.

**2) Internal TTS.** The internal TTS first generates a reasoning trajectory by Long-CoT using $LLM_{\text{think}}$, and then predicts the final answer with this trajectory using $LLM_{\text{answer}}$, which can be expressed as:

$$\text{answer} = LLM_{\text{answer}}(LLM_{\text{think}}(\text{query})). \tag{2}$$

To be specific, recent methods (e.g. DeepSeek R1(Guo et al., 2025)) use the same policy model for both $LLM_{\text{think}}$ and $LLM_{\text{answer}}$.

**3) External TTS.** Firstly, the Long-CoT generation is extended by generating multiple reasoning trajectories and answers in parallel. Then, a reward model (e.g. PRM) is used to score and select the best result (Lightman et al., 2023; Liu et al., 2025). This inference form can be described as:

$$\text{answer} = \underset{i \in [1,k]}{\arg\max} \, LLM_{PRM}\Big( [LLM_{\text{answer}}(LLM_{\text{thinking}}(\text{query}))]_i \Big), \tag{3}$$

where $[*]_i$ denotes the $i$-th candidate among $k$ parallel generations.

Though existing external TTS methods have been proven to obtain considerable performance enhancement, they still encounter several problems: (1) Extra Computation: PRM contains individual parameters from the policy model ($LLM_{\text{think}}$ and $LLM_{\text{answer}}$), which introduces additional huge computation. (2) Expensive Annotation: It is difficult to obtain the large-scale reasoning trajectory annotations for PRM training.

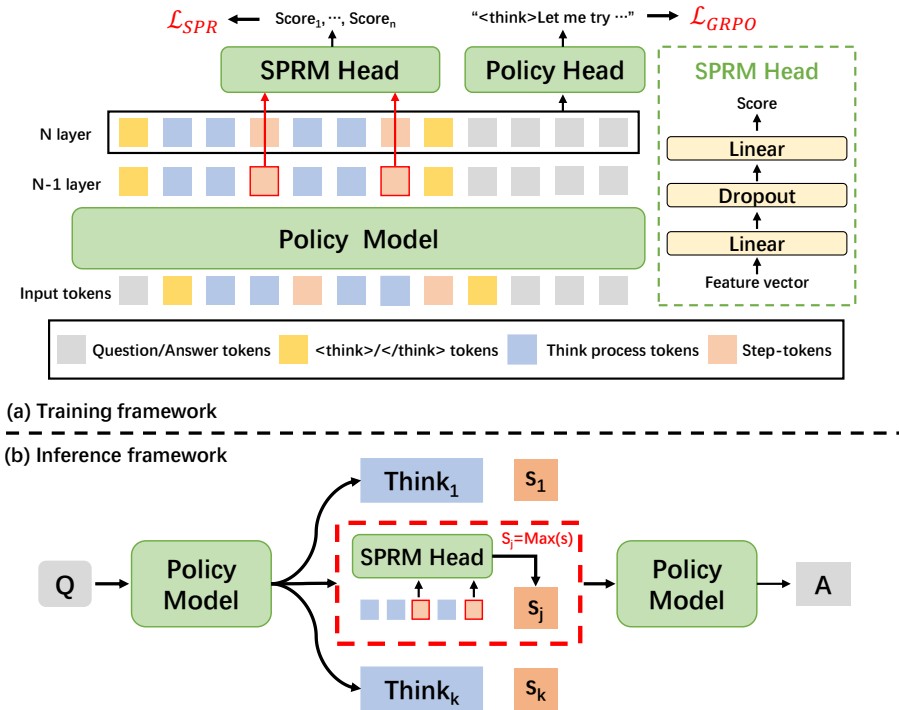

Figure 2: The training and inference framework of Reflective Generative Models.

**Reflective Generative Form.** To address the extra computation and expensive annotation issues, we propose a new Reflective Generative Form focusing on the efficient and label-free reasoning trajectory selection. The proposed Reflective Generative Form is shown in follows,

$$\text{answer} = \underbrace{LLM_{\text{answer}}}_{\text{share backbone}} \Big( \arg\max_{i \in [1,k]} \underbrace{LLM_{SPRM}}_{\text{share backbone}} \Big( [\underbrace{LLM_{\text{thinking}}}_{\text{share backbone}}(\text{query})]_i \Big) \Big) \quad (4)$$

Firstly, we share the backbone of the policy model and PRM in a single network, which enables reasoning trajectory generation and scoring in a unified interface for parallel prediction. The score measures the quality of each reasoning trajectory, and the trajectory with higher score is selected as the high-quality candidate in TTS. This unified interface is proved to be effective for parameter reduction in our experiments. Secondly, we introduce a novel Self-supervised Process Reward Model (SPRM) to eliminate the reliance on process-level annotation, which can be optimized with only outcome-level annotation in a self-supervised manner. In particular, we only implement the SPRM for the $LLM_{\text{think}}$ selection, which can further improve the inference efficiency during the real implementation.

## 4 APPROACH

### 4.1 UNIFIED INTERFACE IN REFLECTIVE GENERATIVE FORM

Our proposed Reflective Generative Form establishes a unified interface for the policy model and the PRM. For the policy model, we employ reasoning LLMs that contain the thinking process in response, delineated by the '<think>' and '</think>' tokens. For the PRM, we introduce a Self-supervised Process Reward Model (SPRM), which shares the same backbone as the policy model but incorporates an additional lightweight SPRM head. The SPRM head is implemented by a binary classifier consisting of two linear layers and a dropout layer: $\text{Linear}(c, 2c) \rightarrow \text{ReLU} \rightarrow \text{Dropout}(0.5) \rightarrow \text{Linear}(2c, 1)$, where $c$ is the channel of the input hidden states. An overview of the joint framework is illustrated in Fig. 2(a).

Within this unified form, the policy model first generates multiple thinking processes as the reasoning trajectories. Subsequently, the SPRM evaluates each thinking process for reasoning trajectory selection. The evaluation procedure contains two steps:

**1. Step Segmentation.** We segment each reasoning trajectory using tokens that are already supported by the policy model's tokenizer, eliminating the need to introduce additional step-specific tokens or fine-tune the LLM for step-format outputs. Specifically, we treat tokens containing '.\n\n' as step-tokens and split the trajectory accordingly. Additionally, we retain only the first token in any sequence of consecutive step-tokens and ignore the step-token appearing at the beginning of the trajectory, as it does not contain valuable information.

**2. Trajectory Score Prediction.** After using step-tokens to mark the end of individual reasoning steps, we evaluate each step based on the representation of the corresponding step-token. Since the representation in the last layer mainly captures the logits prediction for a single token, we use the hidden representations from the second-to-last layer of the policy model to provide richer contextual information of the entire step. These representations are then fed into the SPRM head to predict process scores for each step. When calculating the final score, Lightman et al. (2023) proposes to use the product of process scores. However, this results in lower final scores for longer reasoning trajectories. Thus, we further use the geometric mean of the process scores to eliminate the influence step numbers.

$$S_{\text{final}} = \left(\prod_{n=1}^{N} \text{Score}_n\right)^{\frac{1}{N}} = \left(\prod_{n=1}^{N} SPRM(f_{token_n})\right)^{\frac{1}{N}}, \tag{5}$$

where $N$ denotes the total number of steps, and $f_{\text{token}_n}$ is the representation of the $n$-th step-token obtained from the policy model. $\text{Score}_n$ is the SPRM's process score for $n$-th step.

Through this unified interface, a single network can generate reasoning trajectories and score them in parallel, enabling joint training in an end-to-end manner. This design facilitates a straightforward and efficient training pipeline for on-policy PRM learning, where both the policy model and the SPRM continuously refine their parameters from shared experiences, thereby improving the overall quality of the generated trajectories.

## 4.2 Optimization of Reflective Generative Form

During optimization, we train the policy model and the SPRM head simultaneously. For the policy model, we adopt Group Relative Policy Optimization (GRPO) following Shao et al. (2024). To optimize the SPRM head, we propose a Self-supervised Process Reward Loss (SPR Loss), which enables learning process discrimination ability only from outcome reward (e.g. final answer correctness). The SPR Loss is formulated as follows,

$$\mathcal{L}_{\text{SPR}} = \frac{1}{N} \sum_{n=1}^{N} \mathbb{I}(y = \hat{y_n}) * BCELoss(\text{Score}_n, \hat{y_n}), \quad \text{where } \hat{y_n} = \mathbb{I}(\text{Score}_n > 0.5), \tag{6}$$

where $\mathbb{I}$ is the indicator function, $n$ denotes the step-tokens, $\text{Score}_n$ is SPRM's process score on step $n$, $\hat{y_n}$ is the pseudo label from SPRM on step $n$, and $y$ denotes whether the final answer from the policy model is correct. Since a correct final answer may include incorrect intermediate steps and vice versa (Lightman et al., 2023), we optimize the process score based on both final answer correctness and the pseudo label from SPRM. Specifically, we only update the steps when the pseudo label is consistent with the final answer's correctness. This dynamic filtering allows the model to avoid noisy samples and focus on the most representative steps of correct and incorrect solutions. Thus, by enlarging the score gap between correct and incorrect steps, SPRM can progressively learn the process evaluation ability with only final annotations.

## 4.3 Inference with Reflective Generative Form

In the inference stage, our Reflective Generative Form is naturally suitable for TTS where the SPRM can provide guidance for selecting the high-quality reasoning trajectory from the policy model. The total inference process divides into three steps(shown in Fig. 2(b)): (1) For the given question, the policy model first samples $k$ thinking processes as the candidate reasoning trajectories: $think_1, think_2, \ldots, think_k$. (2) The SPRM evaluates the steps in each process and obtains the final

score by the geometric mean of corresponding process scores: $S_1, S_2, \ldots, S_k$. (3) The reasoning trajectory with the highest final score is chosen and guides the policy model to answer the question (Eq.7).

$$\text{answer} = LLM_{\text{answer}}(think_{i^*}), \text{ where } i^* = argmax(S_1, S_2, \ldots, S_k) \quad (7)$$

### 4.4 DISCUSSION WITH OTHER METHODS

**Comparison with other PRMS.**   Recent works (Rafailov et al.; Chen et al., 2025; Zhong et al., 2024; Yuan et al., 2024) can also generate process reward with only final answers. However, Rafailov et al.; Zhong et al. (2024) require the reference model in Reinforcement Learning to help calculate the reward, which is mainly used for improving the training efficiency of the policy model. Chen et al. (2025); Yuan et al. (2024) require training additional LLM-based reward models (e.g., Llama-3-70B-Instruct in Chen et al. (2025) and ImplicitPRM-8B in Yuan et al. (2024)). Overall, these methods still depend on external LLMs as the reward model, thus assigning the process evaluation capability to an additional model. In comparison, our method unifies the process reward model and the policy model within a single LLM, thereby integrating both reasoning and evaluation capabilities into a single model.

**Comparison with other External TTS methods.**   Recent works Toshniwal et al. (2025); Qi et al. (2025) also focus on External TTS. Toshniwal et al. (2025) integrates both response generation and evaluation into a single LLM by feeding the model's own responses back to itself through manually designed prompts. Qi et al. (2025) trains an additional reward model to jointly evaluate multiple sampled trajectories. However, these approaches still fail to fully unify the generation and evaluation processes. In Toshniwal et al. (2025), it requires manually designed prompts. Besides, its generation and evaluation process are decoupled: the model must be queried multiple times to compare candidate responses, and its evaluation requires an autoregressive reasoning process, which introduces additional computational cost. Meanwhile, Qi et al. (2025) still requires training an additional LLM for evaluation. In contrast to them, our RGM unifies generation and evaluation in both model architecture and inference pipeline, requiring neither additional models nor extra forward passes.

## 5 EXPERIMENT

### 5.1 BASELINE & DATASET

We conduct experiments on the baseline models with different sizes and architectures, including DeepScaleR-1.5B-Preview (Luo et al., 2025), DeepSeek-R1-Distill-Qwen-7B (Guo et al., 2025), QWQ-32B(Team, 2025), Qwen3-32B (Team, 2025), and GPT-OSS-20B. For all models, we add the SPRM head into the second-to-last layer while keeping the remaining architecture unchanged. Our training dataset is sampled from multiple publicly available math-related sources, including NuminaMath (Li et al., 2024), OpenR1-Math-220k, DeepScaleR (Luo et al., 2025), LIMR (Li et al., 2025), and OREAL-RL (Lyu et al., 2025). During data cleaning, first, we use a difficulty classification model trained on the MATH dataset to filter out easy data. Then, we use the pass rate to sample valuable data, following Li et al. (2025). We finally sampled 40k training examples. In the training stage, the models are trained on 64 H200 GPUs with batch size of 128 and response length of 32k. We train the models using GRPO and our proposed SPR loss for 80 iterations (140 steps for QwQ-32B as explained in Sec5.4). In the inference stage, we use the sampling temperature of 1.0 for GPT-OSS-20B and 0.6 for other models. The output length is set to 38k for mathematical tasks and 32k for other tasks. We denote $RGM_k$ for models reasoning with $k$ candidates in Eq.7.

We evaluate our models on 4 challenging mathematical benchmarks: AIME2024/2025 (AIME, 2025), BRUMO25 (Balunović et al., 2025), and HMMT25 (Balunović et al., 2025). To verify the robustness and generalization on other general tasks, we further introduce an extra out-of-distribution benchmark: LivecodeBench(240801-250201) (Jain et al., 2024) (for coding capability evaluation).

Following recent works Zou et al. (2025); Choudhury (2025), we use the results of BoN to test the capability of PRMs. We adopt Pass@1 as the evaluation metric. For each problem, the model

| Model | TTS | Mathematical | | | | Out-of-Distribution |
| --- | --- | --- | --- | --- | --- | --- |
| | | AIME24 | AIME25 | BRUMO25 | HMMT25 | LiveCodeBench |
| *Open-Source Models* | | | | | | |
| **s1-32B** | ✓ | 56.7 | 50.0 | - | - | - |
| **R1-Distill-Qwen-32B** | - | 72.6 | 49.6 | 68.3 | 33.3 | 57.2 |
| **GLM-Z1-32B-0414** | - | 80.8 | 63.6 | - | - | 59.1 |
| **DeepSeek-R1-671B** | - | 79.8 | 70.0 | 80.8 | 44.4 | 65.9 |
| *Closed-Source Models* | | | | | | |
| **Claude-3.5-Sonnet** | - | 16.0 | 7.4 | - | - | 37.2 |
| **GPT-4o-0513** | - | 9.3 | 11.6 | - | - | 32.9 |
| **OpenAI o1-mini** | - | 63.6 | 50.7 | - | - | 53.8 |
| **OpenAI o1-1217** | - | 79.2 | - | - | - | 63.4 |
| **OpenAI o3-mini(med)** | - | 79.6 | 74.8 | 80.0 | 53.0 | 67.4 |
| **DeepScaleR-1.5B** | - | 43.1 | 30.0 | 37.4 | 19.3 | 22.9 |
| **+GRPO** | - | 43.8 | 29.9 | 38.9 | 18.1 | 22.4 |
| **+RM-72B** | ✓ | 50.7 | 33.2 | 41.4 | 16.7 | 24.2 |
| **+PRM-72B** | ✓ | 52.5 | 34.6 | 42.0 | 18.2 | 22.4 |
| **+RGM$_8$-5M** | ✓ | **53.1**(+0.6) | **35.7**(+1.1) | **43.2**(+1.2) | **21.5**(+3.3) | **26.6**(+2.4) |
| **R1-Distill-Qwen-7B** | - | 55.5 | 39.2 | 51.6 | 24.1 | 37.6 |
| **+GRPO** | - | 54.7 | 41.2 | 52.3 | 25.0 | 39.4 |
| **+RM-72B** | ✓ | 56.5 | 43.8 | 55.8 | 27.5 | 41.7 |
| **+PRM-72B** | ✓ | 60.1 | 47.3 | 55.7 | 29.9 | 42.8 |
| **+RGM$_8$-26M** | ✓ | **66.3**(+6.2) | **48.3**(+1.0) | **56.9**(+1.1) | **33.4**(+3.5) | **44.1**(+1.3) |
| **QwQ-32B** | - | 79.5 | 69.5 | 75.2 | 47.5 | 63.4 |
| **+GRPO** | - | 79.9 | 70.5 | 75.8 | 47.1 | 63.4 |
| **+RM-72B** | ✓ | 82.9 | 71.7 | 76.5 | 46.4 | 62.9 |
| **+PRM-72B** | ✓ | 83.3 | 72.3 | 76.5 | 51.7 | 63.0 |
| **+RGM$_8$-54M** | ✓ | **84.2**(+0.9) | **73.4**(+1.1) | **78.1**(+1.6) | **53.1**(+1.4) | **64.0**(+1.0) |
| **Qwen3-32B** | - | 81.4 | 72.9 | 78.0 | 51.9 | 64.1 |
| **+GRPO** | - | 81.7 | 73.3 | 78.3 | 52.1 | 64.6 |
| **+RM-72B** | ✓ | 80.5 | 74.4 | 79.2 | 52.3 | 63.3 |
| **+PRM-72B** | ✓ | 82.9 | 77.1 | 83.5 | **56.7** | 64.9 |
| **+RGM$_8$-54M** | ✓ | **83.8**(+0.9) | **77.6**(+0.5) | **85.8**(+2.3) | 56.3(-0.4) | **66.2**(+1.3) |
| **GPT-OSS-20B(med)** | - | 80.0 | 72.1 | 71.2 | 55.5 | 67.9 |
| **+GRPO** | - | 78.1 | 73.9 | 71.7 | 55.8 | 69.8 |
| **+RM-72B** | ✓ | 80.8 | 77.7 | 71.8 | 60.4 | 68.4 |
| **+PRM-72B** | ✓ | **82.2** | 78.1 | 73.1 | 58.9 | 69.5 |
| **+RGM$_8$-17M** | ✓ | 81.9(-0.3) | **79.1**(+1.0) | **73.3**(+0.2) | **63.4**(+3.0) | **69.9**(+0.4) |

Table 1: Comparison of our RGM and other models. The best results are shown in **bold**. The values in parentheses indicate the performance gain over other TTS methods. Lines start with "RGM$_8$-XM" denotes our method and X denotes the number of parameters (in millions) of our SPRM.

generates only one final answer, and the Pass@1 score is computed as the proportion of correctly solved problems. To improve the stability of the results, we repeat the evaluation 64 times and report the average accuracy as the final score.

## 5.2 Main Results

Table 1 summarizes the performance of our proposed RGM across four representative benchmarks. The baseline is the models trained with only GRPO on our training set. To evaluate the effectiveness of RGM, we compare it with two widely adopted large-scale reward models: Qwen2.5-Math-RM-72B (an outcome reward model trained on 600k math problems) (Yang et al., 2024) and Qwen2.5-Math-PRM-72B (a process reward model trained on 500k math problems) (Zhang et al., 2025). These reward models are also applied to the baseline models with GRPO training. When testing with TTS, we sample 8 candidates for each problem and predict the final answer based on the best candidate using the reward model. As listed in Table 1, the baseline with GRPO has similar results to the basic policy models. This is reasonable as we only trained the policy models with a few iterations. After using RGM, our method consistently surpasses the baseline and the original models by a significant margin, especially on mathematical tasks. This verifies that the main improvement of RGM comes from our proposed SPRM. When comparing with other reward models, despite

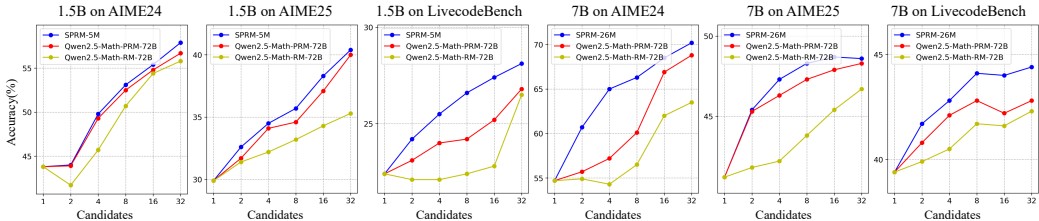

Figure 3: Evaluation of varying numbers of candidate reasoning trajectories.

having only million-level parameters, our RGM achieves comparable or even better performance than billion-level reward models. Specifically, our RGM outperforms other 72B reward models on AIME25, BRUMO25, and LiveCodeBench for all models. On AIME24 and HMMT25, our RGM still surpasses other 72B reward models on most architectures. Finally, we also compare RGM with several advanced open-source models (DeepSeek-R1-Distill-Qwen-32B, GLM-Z1-32B-0414 (GLM et al., 2024), s1-32B (Muennighoff et al., 2025), and DeepSeek-R1-671B (Guo et al., 2025)) and closed-source models (Claude-3.5-Sonnet-1022, GPT-4o-0522, OpenAI o1-mini, OpenAI o1-1217, and OpenAI o3-mini-medium). Our RGM also surpasses all these advanced LLMs, Especially on mathematical tasks, R1-Distill-Qwen-7B with RGM achieves performance comparable to OpenAI o1-mini, and QwQ-32B with RGM achieves performance comparable to OpenAI o3-mini. These demonstrates the ability of our RGM to improve the performance upper bound of advanced models.

In summary, the results demonstrate that: 1) With substantially fewer parameters and less training data, the proposed SPRM achieves superior performance. This highlights the effectiveness of RGM in enhancing reasoning ability without the need for additional large-scale reward models. 2) Although SPRM is trained only on mathematical data, the improvements in reasoning capability generalize to other domains (e.g., Qwen3-32B improves from 64.6 to 66.2 on LiveCodeBench). Appendix A further demonstrates its generalization ability on Chinese tasks. These results confirm that the reasoning gains from RGM are not limited to mathematical tasks. Instead, they generalize robustly to other domains, highlighting RGM's strong transferability.

## 5.3 ABLATION STUDY

**The candidate number in SPRM.** To examine the effect of the number of candidate reasoning trajectories $k$, we report the results for different $k$ in Fig. 3. The results of Qwen2.5-Math-RM-72B and Qwen2.5-Math-PRM-72B are also listed. It is shown that a larger $k$ results in a higher performance. Besides, across different $k$ and model sizes, SPRM consistently outperforms other large scale reward models, indicating its strong ability to distinguish between high and low quality reasoning trajectories. The detailed examples of SPRM are shown in Appendix.E

**Effectiveness of self-supervised optimization.** We evaluate the effectiveness of SPRLoss in Table 2. Compared with using the final answer correctness as process-level supervision for PRM training, our proposed self-supervised optimization method achieves larger performance gains on both 1.5B and 7B models. Furthermore, Fig. 4 shows the prediction score gap between correct and incorrect solutions. Compared to the BCELoss, SPRLoss demonstrates stronger discriminative capability with a larger score gap. This indicates that treating final answer correctness as process-level labels introduces substantial label noise, which harms the optimization. In contrast, SPRLoss leverages self-supervised signals to reduce the impact of noisy supervision, leading to stable and accurate training.

**The calculation of final score.** In Eq. 5, we compute the final score using the geometric mean of the process scores. In Table 3, we compare the geometric mean with two alternatives: the arithmetic mean and the direct product of scores. The results show that the geometric mean achieves the best performance, , and there is a large gap between the product and the geometric mean of process scores. This is expected, as for long reasoning trajectories, the score tends to diminish as the number of steps increases, causing the final score to be overly sensitive to the response length. In contrast, the geometric mean alleviates this bias by reducing the impact of trajectory length.

| Model | Loss | AIME24 | LiveCodeBench |
|---|---|---|---|
| DeepScaleR-1.5B+RGM$_{32}$ | BCELoss | 56.7 | 27.9 |
| | SPRLoss | **57.9** | **28.1** |
| R1-Distill-Qwen-7B+RGM$_{32}$ | BCELoss | 69.1 | 43.9 |
| | SPRLoss | **70.2** | **44.4** |

Table 2: Evaluation on SPRLoss.

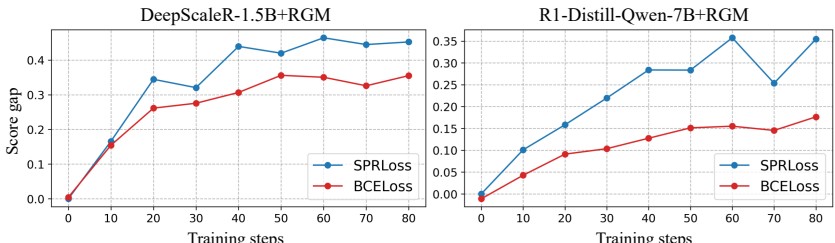

Figure 4: The prediction score gap between correct and incorrect solutions. The blue curve shows the SPRLoss. The red curve shows the BCELoss.

| Method | AIME24 | AIME25 | BRUMO25 | HMMT25 |
|---|---|---|---|---|
| Product | 44.2 | 31.1 | 40.0 | 17.9 |
| Arithmetic mean | 52.9 | 35.2 | **43.3** | 21.1 |
| Geometric mean | **53.1** | **35.7** | 43.2 | **21.5** |

Table 3: Evaluation on the calculation of final score on DeepScaleR-1.5B+RGM$_8$..

## 5.4 AHA MOMENT OF RGMS

Guo et al. (2025); Hu et al. (2025) propose that the "aha moment" enables the model to perform self-correction and self-reflection. In RGM, as we propose an SPRM head to evaluate itself, we define an "aha moment" as the step at which the SPRM starts to discriminate the correct and incorrect reasoning trajectories. In Fig. 5, we present curves of the final evaluation scores from RGM for correct and incorrect reasoning trajectories. At the initial phase of training, due to the prediction bias of the initial model, the scores for both correct and incorrect samples increase rapidly, indicating that the SPRM is dominated by the correct samples. Besides, the pseudo labels of the initial model also contain noisy interference for optimization. However, we observe a step at which the optimization behaviors for correct and incorrect reasoning trajectories begin to diverge, indicating that the SPRM starts to acquire the ability to discriminate — the "aha moment". Formally, the "aha moment" is defined as the first training step at which the slope of the curve for correct trajectories becomes positive while that for incorrect trajectories becomes negative.

This observation indicates that, under our unified training framework and self-supervised loss, the bias issue can be mitigated. After this "aha moment", RGM iteratively learns to evaluate itself with SPRLoss, resulting in a clear distinction between correct and incorrect reasoning trajectories and thereby enabling effective external TTS. The typical "aha moment" case study can be found in Appendix.E. Specifically, since the "aha moment" for QwQ-32B occurs around 60 steps, we further train the model for additional 60 steps to ensure model convergence.

## 5.5 PROCESS-LEVEL EVALUATION OF RGMS

To evaluate whether the improvements of RGMs come from process-level rewards, we conduct an additional experiment in which only the score of the final step token is used (treating SPRM as an outcome reward model). As shown in Table 4, incorporating step-level tokens within the reasoning trajectory leads to notable improvements in final performance (4.6/0.7/0.7/2.7/0.9

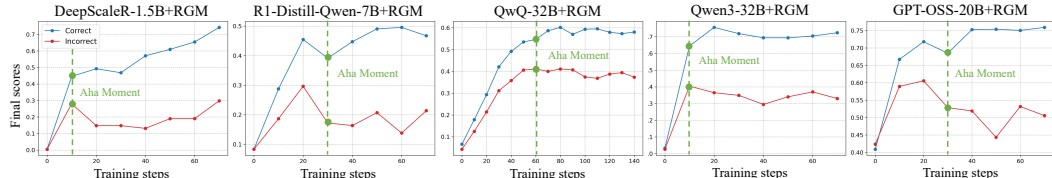

Figure 5: The training process of SPRM. The blue and red curves denote the final score on correct and incorrect reasoning trajectories. The green dashed line indicates the "aha moment".

on AIME24/AIME25/BRUMO25/HMMT25/LiveCodeBench). This highlights the importance of process-level rewards in guiding reasoning.

| Method | AIME24 | AIME25 | BRUMO25 | HMMT25 | LiveCodeBench |
|---|---|---|---|---|---|
| **Outcome-level** | 48.5 | 35.0 | 42.5 | 18.8 | 25.7 |
| **Process-level** | **53.1** | **35.7** | **43.2** | **21.5** | **26.6** |

Table 4: Performance of DeepScaleR-1.5B+RGM$_8$.

Furthermore, we introduce a process-level TTS method based on Monte Carlo Tree Search (MCTS) for evaluation. MCTS requires the reward model to select intermediate steps during the reasoning process, rather than only evaluating the whole reasoning trajectory. The inference settings for this experiment are provided in Appendix C. In Table 5, increasing the maximum number of searching tokens to means more intermediate steps are selected by our SPRM. And the performance consistently improves from 43.8 to 52.8 with increasing searching tokens. These results demonstrate the effectiveness of SPRM in providing high-quality process-level guidance.

| Searching Tokens(k) | 0 | 40 | 80 | 120 | 160 |
|---|---|---|---|---|---|
| **Accuracy(%)** | 43.8 | 48.8 | 50.0 | 51.7 | 52.8 |

Table 5: Performance of DeepScaleR-1.5B+RGM with MCTS on AIME24. A larger searching token number indicate more process steps are selected by our SPRM.

However, the performance of MCTS remains below that of the Best-of-N strategy reported in Table 1. We attribute this gap primarily to the computational overhead of tree-based search, which leads to incomplete exploration under our experimental settings. Specifically, long reasoning trajectories in challenging tasks correspond to deep search levels in MCTS. This results in a very large search space, as the total number of explored nodes grows rapidly with increasing depth. Consequently, the computational overhead of MCTS becomes extremely high. Under a limited compute budget, stopping the search at an early stage not only restricts the search space but also risks missing errors that appear in later steps of the reasoning trajectory, resulting in degraded performance. Nevertheless, the observed gains over the baseline confirm the capability of our RGMs to identify and search for better reasoning processes.

## 6 CONCLUSION

In this paper, we propose a novel Reflective Generative Form, which enables a single LLM to both generate and select high-quality reasoning trajectories for Test-Time Scaling (TTS). Based on this form, we present the reflective generative model (RGM). Specifically, we design a unified interface that integrates the policy model and process reward model (PRM) within a single network, resulting in low parameter overhead and efficient TTS inference. A self-supervised process reward model (SPRM) is proposed to learn process-level evaluation with only final answer annotations. Experiments show our QwQ-32B equipped with our RGM reaches comparable performance to OpenAI o3-mini and our SPRM with million level parameters outperforms billion level reward models across most tasks and models.

**Acknowledgment.** This work is supported by the National Nature Science Foundation of China (62425114, 62121002, U23B2028), and the Fundamental and Interdisciplinary Disciplines Breakthrough Plan of the Ministry of Education of China (JYB2025XDXM103).

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

## A    EXTEND ON CHINESE TASKS

We adopt C-Eval (Huang et al., 2023) to evaluate the Chinese QA capability of our RGM. During inference, all tokens containing '\n' are treated as step-tokens. As shown in Table 6, although RGM is trained on English data, it consistently improves Chinese QA performance across all model architectures. These results suggest that RGM provides stable gains without causing performance degradation on unseen tasks, thereby demonstrating its applicability to more general scenarios.

| Model | DeepScaleR-1.5B | R1-Distill-Qwen-7B | QwQ-32B | Qwen3-32B | GPT-OSS-20B(med) |
|---|---|---|---|---|---|
| **Baseline** | 39.9 | 56.8 | 89.4 | 89.0 | 67.8 |
| **+RGM$_8$** | **43.9** | **62.5** | **89.6** | **89.5** | **68.1** |

Table 6: Performance of RGM on C-Eval.

## B    ADDITIONAL ABLATION STUDIES

**The threshold in SPRLoss.**    In Eq. 6, we adopt a hard threshold of 0.5 to obtain pseudo labels. Here, we further compare two dynamic alternatives that use the sample-wise mean and median as the threshold. The results are reported in Table 7. All three thresholding strategies achieve comparable performance, while the hard threshold performs slightly better. We attribute this to the fact that a fixed hard threshold provides a sample-independent and absolute criterion for pseudo-label partitioning, enabling the predicted scores to be comparable across different samples.

| Method | AIME24 | AIME25 | BRUMO25 | HMMT25 |
|---|---|---|---|---|
| Mean | 50.0 | 35.0 | **45.0** | 21.1 |
| Median | 51.7 | 35.0 | 42.5 | 20.0 |
| Hard | **53.1** | **35.7** | 43.2 | **21.5** |

Table 7: Evaluation on the threshold in SPRLoss on DeepScaleR-1.5B+RGM$_8$.

**Independent reward model.**    In Table 8, we train an external outcome reward model and an RGM without parameter sharing, as shown in Table 8. Compared with the outcome reward model, our RGM has smaller total parameters (share backbone), simpler training pipeline (one stage training), and higher performance.

As for the independent RGM, the results indicate that the shared-parameter and external models achieve comparable performance improvement compared with the baseline (around 1% accuracy difference). This slight bias is reasonable as we combine 2 tasks in a single model. Since the independent reward model also leads to huge training cost (training 2 LLMs) and parameter cost(2X parameter cost than RGM), This does not affect our main conclusion that our method offers a lightweight and effective approach for External TTS.

| Method | Extra Param | AIME24 | AIME25 | BRUMO25 | HMMT25 |
|---|---|---|---|---|---|
| Deepscaler-1.5B | - | 43.1 | 30.0 | 37.4 | 19.3 |
| +Independent ORM | 1.5B | 51.7 | 33.3 | 41.7 | 18.9 |
| +Independent RGM$_8$ | 1.5B | 54.7 | 36.7 | 43.8 | 21.9 |
| +RGM$_8$ | 5M | 53.1 | 35.7 | 43.2 | 21.5 |

Table 8: Evaluation on independent reward models on DeepScaleR-1.5B+RGM$_8$.

**The position of SPRM.**    We add the SPRM to the second-to-last layer of the policy model. To further examine the effect of its placement, Table 9 compares it with attaching the SPRM to the final layer. The results showed that using the second-to-last layer can obtain better performance. This behavior is expected. First, the final layer must directly compute similarity with the policy

classifier kernels for prediction, which limits its ability to retain contextual information. Second, using a single feature representation for two different classifiers introduces task interference, whereas intermediate-layer features are more suitable for auxiliary objectives.

| Method | AIME24 | AIME25 | BRUMO25 | HMMT25 |
|---|---|---|---|---|
| Last layer | 48.8 | 34.6 | 41.3 | 18.8 |
| Second-to-last layer | **53.1** | **35.7** | **43.2** | **21.5** |

Table 9: Evaluation on the position of SPRM on DeepScaleR-1.5B+$RGM_8$..

**Comparison with majority.**   Table 10 further shows the comparison with majority voting. We can see that our SPRM obtains better performance. Besides, our RGM is more flexible. For some general scenarios (e.g., code generation), simple majority voting may not be applicable.

| Method | AIME24 | AIME25 | BRUMO25 | HMMT25 |
|---|---|---|---|---|
| Majority | 50.4 | 34.0 | 41.7 | 20.3 |
| RGM | **53.1** | **35.7** | **43.2** | **21.5** |

Table 10: Comparison with majority voting on DeepScaleR-1.5B+$RGM_8$..

**The model design.**   In this paper, we implement SPRM using two linear layers. Table 11 further analyzes different SPRM architectures: a single linear layer and a two-layer model with an additional gating mechanism. The results indicate that a single linear layer is insufficient for reliable evaluation, whereas a more expressive architecture yields better performance.

| Method | AIME24 | AIME25 | BRUMO25 | HMMT25 |
|---|---|---|---|---|
| Linear | 48.5 | 35.5 | 38.9 | 20.0 |
| Linear*2 + Gate | **53.3** | 35.6 | **46.7** | **22.3** |
| Linear*2 | 53.1 | **35.7** | 43.2 | 21.5 |

Table 11: Evaluation with different architectures of SPRM on DeepScaleR-1.5B+$RGM_8$..

**Comparison with ImplicitPRM.**   The ImplicitPRM (Yuan et al., 2024) also does not require process annotations. However, it still depends on at least one additional reward model (the externally trained 8B ORM) during prediction. In Table 12, we compare our RGM with ImplicitPRM-8B. The results show that our RGM outperforms it while using fewer extra parameters. Moreover, similar to previous PRMs, the ImplicitPRM requires a two-stage training and inference pipeline, while our RGM unifies the process reward model and the policy model within a single LLM, enabling efficient test time scaling.

| Method | Extra Param | AIME24 | AIME25 | BRUMO25 | HMMT25 |
|---|---|---|---|---|---|
| Deepscaler-1.5B | - | 43.1 | 30.0 | 37.4 | 19.3 |
| +ImplicitPRM | 8B | 52.5 | 35.5 | 40.8 | 20.0 |
| +$RGM_8$ | 5M | **53.1** | **35.7** | **43.2** | **21.5** |

Table 12: Comparison with ImplicitPRM on DeepScaleR-1.5B+$RGM_8$..

## C   DETAILS FOR MCTS

In the expanding stage, we expand 4 children for the selected node. Since the complete reasoning trajectory is very long in challenging benchmarks, we generate 1024 tokens as 1 step in each node to reduce the complexity of MCTS. Instead of performing full simulations to the end of a reasoning

trajectory, we directly use SPRM to estimate the value of current node during the searching stage. This enables a more efficient evaluation at each step and is more suitable for evaluating the process reward of our SPRM. To balance the computation cost, we set the maximum number of total searching tokens during the MCTS process from 0 (without MCTS) to 160k for each question.

## D DISCUSSION

**Dependency of SPRM on Policy Models.** Compared with existing pretrained reward models, our proposed SPRM requires training on the target policy models, which introduces an additional training process when adapting to new policy models. However, we argue that such a design is in fact necessary for building reliable reward models. Traditional pretrained reward models can indeed be directly applied to unseen policy models, but they inevitably face the risk of out-of-distribution mismatch when the output patterns of the target model deviate from those observed in the reward model's training data. For example, applying RM-72B on Qwen3-32B leads to a performance drop of –1.2 on AIME24; applying PRM-72B on QwQ-32B results in a –1.1 performance drop on HMMT25. To obtain better results, they still need to be finetuned to align the target model, which results in a huge cost for training these large-scale LLMs. In contrast, our SPRM can be trained together when optimizing the policy models, which actually results in no additional training stages. Furthermore, SPRM adopts a lightweight architecture that converges within approximately 100 iterations. Thus, the training cost of SPRM is substantially lower than the optimization of the policy model itself, making it both practical and efficient.

**Transfer of SPRM.** Since different LLMs have different hidden-state dimensions, our SPRM cannot be directly applied to a new model. To address this issue, we add a linear projection layer before the SPRM head to align the feature dimensions, and we only finetune the lightweight SPRM module. In Table 13, we transfer an SPRM trained on Qwen3-32B to DeepScaleR-1.5B. The results show that, without requiring any training of the policy model, the SPRM learned from one model can be effectively transferred to another.

| Method | AIME24 | AIME25 | BRUMO25 | HMMT25 |
|---|---|---|---|---|
| Deepscaler-1.5B | 43.1 | 30.0 | 37.4 | 19.3 |
| +SPRM | 53.1 | 35.7 | 43.2 | 21.5 |
| +SPRM$_{from32B}$ | 52.2 | 37.1 | 45.5 | 21.1 |

Table 13: Evaluation on transferring SPRM from Qwen3-32B to DeepScaleR-1.5B.

**Limitation.** First, despite the promising performance, our method requires the policy model to first generate k complete reasoning trajectories, which can introduce latency. This is a common problem for External TTS methods. However, unlike other External TTS methods that typically set k from 64 to 512, we only set k=8 in most experiments, which makes it easy to obtain results through batch-parallel inference and thereby ensures latency comparable to the original model. Second, for tasks that do not require reasoning or require very little reasoning (e.g., real-world fact-based QA or simple questions), our RGM is less effective due to the lack of reasoning rajectories and step tokens. However, in these cases, we believe the main bottleneck is the LLM's knowledge rather than TTS.

## E CASE STUDY

**Case study of aha moment.** We show an example in Fig. 6. We fix the reasoning trajectory and use RGM before and after the aha moment for scoring. In this case, the model mistakenly confuses $\varepsilon_k$ and $\varepsilon_{k+1}$, resulting in an incorrect solution. Our model fails to recognize the error before the aha moment, while the model after the aha moment can correctly discriminate it.

**Case study of SPRM** Fig. 7 shows the visualization of step-wise evaluation scores from SPRM. It can be observed that SPRM effectively identifies low-quality processes generated by the policy model, including logical error (e.g. the misunderstanding of b in step 58 of example 1) and calculation error (e.g., the incorrect computation $9 \times 21 + 7 = 193$ in step 32 of example 2). SPRM assigns

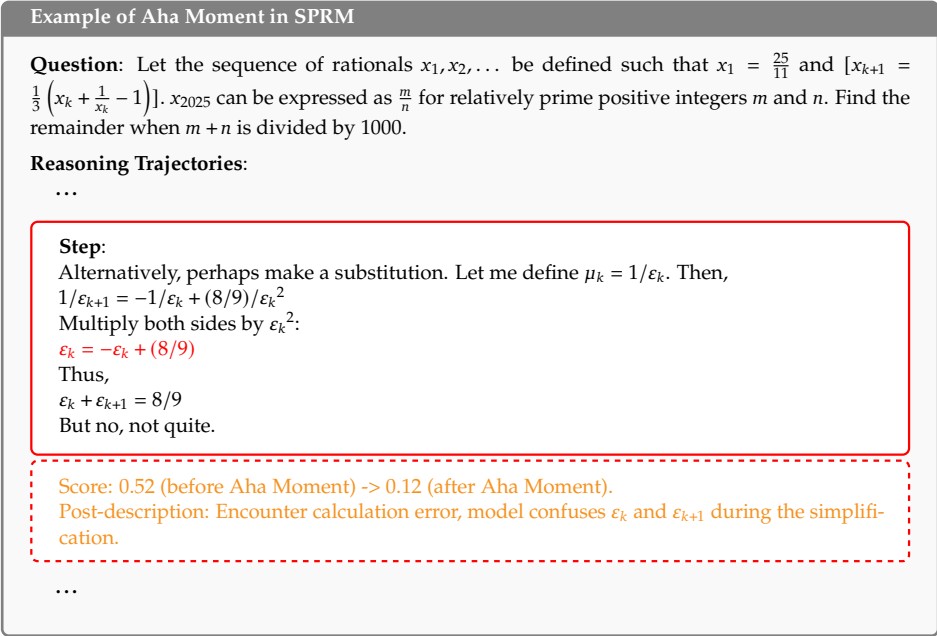

**Example of Aha Moment in SPRM**

**Question**: Let the sequence of rationals $x_1, x_2, \ldots$ be defined such that $x_1 = \frac{25}{11}$ and $[x_{k+1} = \frac{1}{3}\left(x_k + \frac{1}{x_k} - 1\right)]$. $x_{2025}$ can be expressed as $\frac{m}{n}$ for relatively prime positive integers $m$ and $n$. Find the remainder when $m + n$ is divided by 1000.

**Reasoning Trajectories**:

$\cdots$

**Step**:
Alternatively, perhaps make a substitution. Let me define $\mu_k = 1/\varepsilon_k$. Then,
$1/\varepsilon_{k+1} = -1/\varepsilon_k + (8/9)/\varepsilon_k^2$
Multiply both sides by $\varepsilon_k^2$:
$\varepsilon_k = -\varepsilon_k + (8/9)$
Thus,
$\varepsilon_k + \varepsilon_{k+1} = 8/9$
But no, not quite.

Score: 0.52 (before Aha Moment) -> 0.12 (after Aha Moment).
Post-description: Encounter calculation error, model confuses $\varepsilon_k$ and $\varepsilon_{k+1}$ during the simplification.

$\cdots$

Figure 6: Comparison of SPRM's predictions before and after the aha moment. Only key steps are listed. The error steps are marked in red.

low scores to these low-quality steps. Since SPRM only outputs process scores, we additionally provide post-descriptions within the dashed boxes for better clarity.

## F  LLM USAGE STATEMENT

We declare that the LLM was only used as a general-purpose writing assistant to improve the grammar of the manuscript. The LLM did not contribute to the research ideation, methodology design, experimental execution, data analysis, or result interpretation.

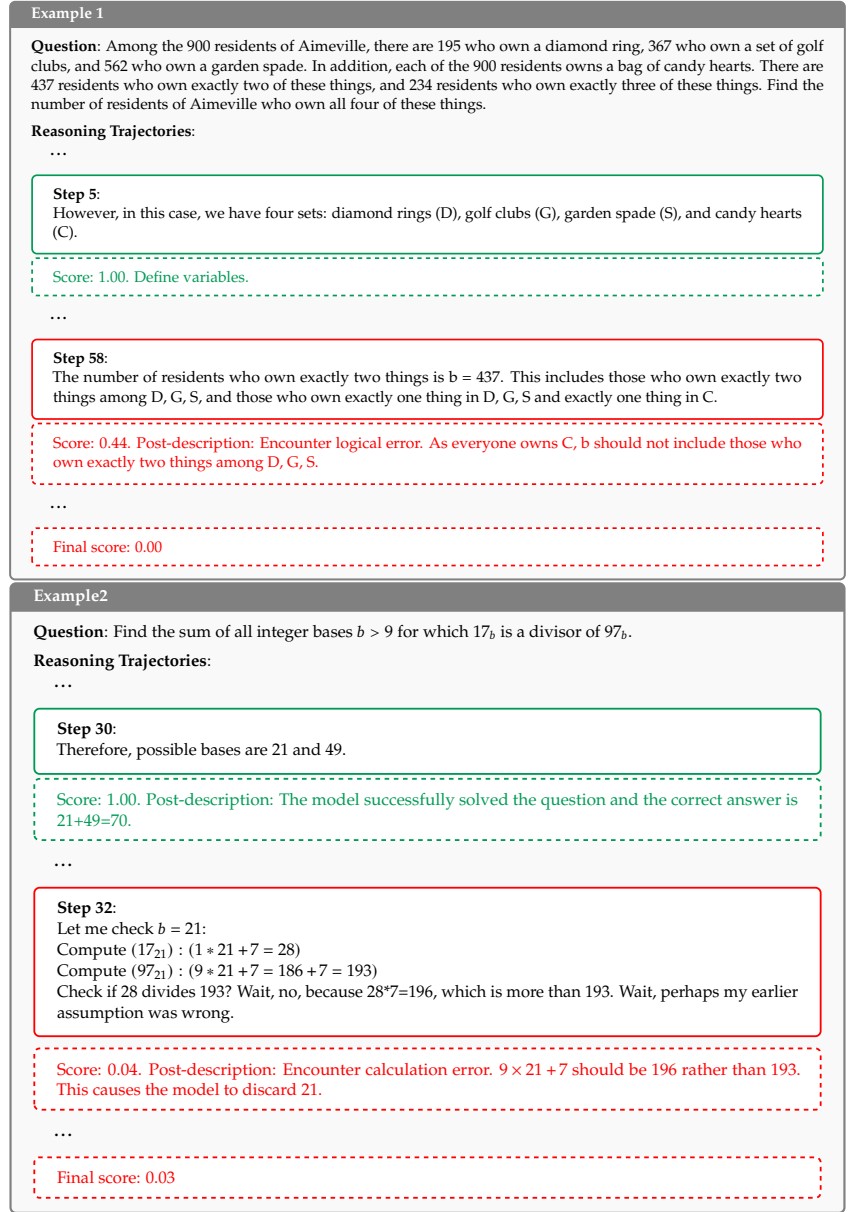

Figure 7: SPRM's predictions on reasoning trajectories. Only key steps are listed. Correct and error steps are marked in green and red.

