# OpenReview forum: "Test-Time Scaling with Reflective Generative Model"
_ICLR.cc/2026/Conference — ICLR 2026 Poster_

### Official Review · Reviewer_uFte · 2025-10-28

**Soundness:** 3
**Presentation:** 3
**Contribution:** 2
**Rating:** 6
**Confidence:** 3

**Summary:**

This paper focuses on parallel TTS techniques to enhance the reasoning capabilities of large models. The authors propose Reflective Generative Model (RGM), where the core idea is to have the policy model and the PRM share the same backbone network. This reduces the parameter and computational overhead associated with deploying a separate PRM. Experiments demonstrate that a 32B model equipped with just a 50M-parameter SPRM can outperform OpenAI's o3-mini on challenging reasoning benchmarks.

**Strengths:**

1.  clear performance improvement is observed as the number of candidates increases, proving the method's effectiveness.
2. The shared-backbone design adds minimal parameter overhead, significantly reducing computational costs.
3. As claimed by the authors, RGM enables efficient reasoning selection while eliminating the need for expensive process-level annotation.

**Weaknesses:**

1. Scalability Concerns: Like many PRM methods, this approach requires manually decomposing responses into "Steps." This process relies heavily on human priors and empirical design, raising doubts about its scalability to new tasks.
2. A core claimed contribution of the paper is removing reliance on process-level supervision. However, ImplicitPRM already claims they can fulfill this (learn PRM from ORM ). How do you position your work relative to this line of work, which suggests that a PRM can function without any extra parameters?

**Questions:**

Could you elaborate on how the "Step-tokens" are identified and inserted during the reasoning process?

Is it possible to compare with the ImplicitPRM methods?

---

> ### Author Response · Authors · 2025-11-26
>
> Dear reviewer,\
> $\qquad$Thank you for your constructive comments. Below, we will respond to weaknesses and questions point by point.
>
> **Weaknesses**:
>
> 1. Q: Scalability concerns on step segmentation
>
> $\quad$R: We thank the reviewer for the comment. Indeed, reliance on step segmentation is a common issue in PRMs. In our work, we use ".\n\n" as a heuristic marker, which appears consistently in long CoT reasoning outputs and Qwen2.5-Math-PRM also uses similar mark (\n\n). While this may not perfectly align with semantic step boundaries, it allows fully automatic segmentation and is effective in new tasks, as shown on code tasks (Table 1) and Chinese tasks (Table 6). In contrast, designing other complex step-segmentation algorithms would introduce strong human priors, potentially reducing generalizability.
>
> 2. Q: Compare with ImplicitPRM
>
> $\quad$R: We have added the discussion with other PRMs (including ImplicitPRM) in Sec4.4.
>
> $\qquad$Specifically, while ImplicitPRM does not require process annotations, it still relies on an externally trained ORM to generate token-level process rewards, assigning process evaluation to an additional model. In contrast, our method unifies the process reward model and the policy model within a single LLM, enabling the model itself to perform reasoning and evaluation without extra models or forward passes.
>
> $\qquad$Besides, SPRM outperforms ImplicitPRM while adding only 5M parameters compared to 8B in ImplicitPRM:
> |Method             | AIME24 | AIME25 | BRUMO25 | HMMT25 |
> | -----             |  ----  |  ----  |  ----   |  ----  |
> | Deepscaler        |  43.1  |  30.0  |  37.4   |  19.3  |
> | ImplicitPRM-8B    |  52.5  |  35.5  |  40.8   |  20.0  |
> | SPRM-5M           |  53.1  |  35.7  |  43.2   |  21.5  |
>
> $\qquad$We have added this ablation study in the paper.
>
> **Questions**:
>
> 1. Q: Details for step-tokens
>
> $\quad$R: Step tokens are not manually inserted. Instead, we directly use the positions of ".\n\n" tokens already generated by the model in its response. Specifically, we record the token IDs corresponding to all ".\n\n" tokens to create a step-token ID list. During inference, whenever the model predicts a token in this list, its hidden representation is fed into SPRM to compute the step score.
>
> 2. Q: Compare with ImplicitPRM
>
> $\quad$R: See the Weakness.2

---

### Official Review · Reviewer_65ZK · 2025-11-03

**Soundness:** 3
**Presentation:** 3
**Contribution:** 3
**Rating:** 6
**Confidence:** 4

**Summary:**

The paper introduces Reflective Generative Models to enable test time scaling. Authors propose a unified interface where the policy model and PRM share the same backbone network, which saves parameter; they also introduce a Self-supervised PRM that learns to evaluate reasoning quality from only outcome-level supervision. Experiment results show that RGM with SPRM achieves performance comparable to OpenAI o3-mini on math benchmarks.

**Strengths:**

* The unified interface design is elegant and addresses a real deployment challenge in test-time scaling. Achieving comparable performance to 72B parameter reward models with only 50M parameters is a solid contribution
* The evaluation is thorough and well-designed with multiple base models and diverse architectures
* Paper writing is clear and readable. The formalization of different inference paradigms provides a clean framework for understanding the contribution
* The "aha moment" analysis is very interesting and provides valuable insights for future works

**Weaknesses:**

* The SPR loss introduces a bootstrapping process where the current model quality decides the psudolabel quality. I am curious if there is any formal analysis on the convergence properties.
* Since the authors are proposing a novel architecture, I would expect more ablation studies on model design, for example the SPRM head MLP structure, and the geometric mean aggregation.
* The method relies on '.\n\n' tokens as semantic boundaries, which assumes the policy model naturally produces well-segmented reasoning. This heuristic may not transfer to models with different output conventions, non-English languages, or domains where reasoning doesn't follow paragraph-like structure.

**Questions:**

* Can the authors provide some analysis on the model's robustness to initialization?
* Is there a way to detect the aha moment online during training, or can you predict when it will occur based on model size or other factors?
* Additional ablation studies or explanations of design choices regarding the model architecture would strengthen the paper. See weakness.
* Given the dependency on training SPRM jointly with each policy model, have you explored whether SPRM trained on one model can transfer to similar architectures with fine-tuning? This would make the approach more practical for people who want to apply it to other base models without access to full retraining.

---

> ### Author Response · Authors · 2025-11-26
>
> Dear reviewer,\
> $\qquad$Thank you for your constructive comments. Below, we will respond to weaknesses and questions point by point.
>
> **Weaknesses**:
>
> 1. Q: Analysis on SPRLoss
>
> $\quad$R: SPRLoss is designed to stabilize learning via confidence-based filtering (Eq. 6). By focusing on steps with high-confidence pseudo labels, the model avoids errors from incorrect intermediate steps, reducing conflicts and enabling stable training. Using the model's own high-confidence predictions as pseudo-labels and iteratively refining the model itself has been verified as an effective method in self-training and semi-supervised training[1][2].
>
> $\qquad$If SPRM initially predicted low (or high) scores for all steps, all positive (or negative) samples would be filtered out, causing the model to only learn negative (or positive) labels and leading to training failure. However, in practice, although the model initially predicts mostly low scores after initialization, there are still a small number of high-score predictions that ensure normal learning (All models in Fig.5 converged).
>
> $\qquad$[1] How does unlabeled data improve generalization in self‑training? ICLR2022
>
> $\qquad$[2] Dash: Semi-supervised learning with dynamic thresholding. ICML2021
>
> 2. Q: Ablation studies on model design
>
> $\quad$R: Thanks for your suggestions. We have added the ablation studies. Results are shown as follows:
>
> $\qquad$ (1) SPRM head. We compare a single FC layer, two FC layers, and two FC layers with a gate. A more expressive head generally improves performance. However, the main contribution of RGM is the unified reasoning-evaluation framework; the two-layer SPRM head used in the paper already demonstrates its effectiveness.
> |Method      | AIME24 | AIME25 | BRUMO25 | HMMT25 |
> | -----      |  ----  |  ----  |  ----   |  ----  |
> | FC         |  48.5  |  35.5  |  38.9   |  20.0  |
> | FC*2 + Gate|  53.3  |  35.6  |  46.7   |  22.3  |
> | FC*2       |  53.1  |  35.7  |  43.2   |  21.5  |
>
> $\qquad$ (2) Geometric mean aggregation. We compare arithmetic mean and product of step scores. Geometric mean balances penalizing low-scoring steps while avoiding extreme reduction for long trajectories, which results in slightly better overall performance.
> |Method           | AIME24 | AIME25 | BRUMO25 | HMMT25 |
> | -----           |  ----  |  ----  |  ----   |  ----  |
> | Product         |  44.2  |  31.1  |  40.0   |  17.9  |
> | Arithmetic mean |  52.9  |  35.2  |  43.3   |  21.1  |
> | Geometric mean  |  53.1  |  35.7  |  43.2   |  21.5  |
>
> 3. Q: Response without step-tokens
>
> $\quad$R: (1) For long CoT reasoning. In our experiments, the LLMs produce long CoT outputs where ".\n\n" often appears and separates paragraph-level steps. Other PRMs such the Qwen2.5-math-PRM also use similar step-token (e.g., "\n\n")
>
> $\qquad$(2) For non-English or short outputs. For Chinese tasks (Appendix A), we use all tokens containing "\n" as step boundaries, which works well in practice. For responses that do not contain ".\n\n", they are typically short answers requiring minimal reasoning; in these cases, TTS scoring is unnecessary.
>
> **Questions**:
>
> 1. Q: Analysis of the model's robustness to initialization
>
> $\quad$R: See the weakness.1
>
> 2.  The way to detect the aha moment
>
> $\quad$R: Formally,  the "aha moment" is defined as the first training step at which the slope of the curve for correct trajectories becomes positive while that for incorrect trajectories becomes negative. This allows online detection by tracking prediction scores for correct and incorrect samples during training.
>
> $\qquad$Currently, the aha moment is an empirical phenomenon used for visualization to show that SPRM begins to reliably discriminate between correct and incorrect steps, indicating stable pseudo-label generation. Precise prediction of its timing based on model size or other factors remains future work.
>
> 3. Q: Ablation studies on model design
>
> $\quad$R: See weakness.2
>
> 4. Q: Transfer SPRM to other LLMs
>
> $\quad$R: Directly applying an SPRM trained on one LLM to another is not possible due to differences in hidden-state dimensions. To enable transfer, we add a linear projection layer to align features and only fine-tune the lightweight SPRM module. As shown below, SPRM trained on 32B model can be effectively transferred to another 1.5B Deepscaler with light fine-tuning, achieving performance close to SPRM trained from scratch:
> |Method             | AIME24 | AIME25 | BRUMO25 | HMMT25 |
> | -----             |  ----  |  ----  |  ----   |  ----  |
> | Deepscaler        |  43.1  |  30.0  |  37.4   |  19.3  |
> | SPRM              |  53.1  |  35.7  |  43.2   |  21.5  |
> | SPRM from 32B     |  52.2  |  37.1  |  45.5   |  21.1  |
>
> $\qquad$This demonstrates that SPRM is lightweight and practically transferable to similar architectures, reducing the need for full retraining.

---

### Official Review · Reviewer_RgpY · 2025-11-04

**Soundness:** 2
**Presentation:** 2
**Contribution:** 2
**Rating:** 4
**Confidence:** 4

**Summary:**

This paper proposes RGM (Reflective Generative Model), a test-time scaling approach that shares the backbone network between the policy model and process reward model (PRM), adding only 50M parameters for trajectory scoring. The key innovation is a Self-supervised Process Reward Model (SPRM) that learns to evaluate reasoning steps using only outcome-level supervision. Experiments show strong empirical results across models of different sizes.

**Strengths:**

- Model overlap in the policy and reward model means that the inference overhead is minimal.
- SPRM can be trained with just the outcome labels, and from the ablation experiments seems to correspond to process-level correctness. For example, score of the last step performs worse than the score of the entire sequence.
- Good empirical results
- Generalization beyond math to coding
- The experimental setup is thorough and tested across multiple model sizes.

**Weaknesses:**

- Important baselines such as majority voting is missing. Moreover, there have been recent work such as "GenSelect: A Generative Approach to Best-of-N" (Toshniwal, 2025) and "Learning to Reason Across Parallel Samples for LLM Reasoning" (Qi, 2025) which demonstrate strong parallel reasoning performance.
- The claims about beating 72B RM is somewhat misleading. The Qwen2.5-RM models were trained on short, non-reasoning CoT solutions, and are not suitable for scoring the long reasoning traces generated by models evaluated in this paper. The parameter count claim of 50M beating 72B RM in the abstract is also somewhat misleading because the RM is sharing the backbone.
- MCTS results being slightly worse than Best-of-N suggests the RM is still not good enough to conduct search which begs the question if a process reward model is really buying any performance in this setup.  Ideally, an ORM trained on the same data as SPRM would be a more fair comparison.

**Questions:**

- Is the "Aha moment" surprising? Isn't it just training dynamics?
- Why did you not train a new ORM on the training data for SPRM?
- Why are obvious baselines like majority voting/self-consistency missing from the paper?

---

> ### Author Response · Authors · 2025-11-26
>
> Dear reviewer,\
> $\qquad$Thank you for your constructive comments. Below, we will respond to weaknesses and questions point by point.
>
> **Weaknesses**:
>
> 1. Q: Comparison with majority voting, similar external TTS works
>
> $\quad$R: We thank the reviewer for the suggestion.
>
> $\qquad$(1) Considering some general scenarios (e.g., code generation), simple majority voting may not be applicable; therefore, we did not include it. Here, we show the comparison of majority voting.
> |Method    | AIME24 | AIME25 | BRUMO25 | HMMT25 |
> | -----    |  ----  |  ----  |  ----   |  ----  |
> | majority |  50.4  |  34.0  |  41.7   |  20.3  |
> | RGM      |  53.1  |  35.7  |  43.2   |  21.5  |
>
> $\qquad$(2) We have added a discussion section in the paper. Specifically, [1–2] also focus on external test-time scaling (TTS). [1] performs evaluation via additional autoregressive reasoning, while [2] trains an additional LLM-based evaluator.
>
> $\qquad$However, these approaches still fail to fully unify the reasoning and evaluation processes. In [1], it requires manually designed prompts. Besides, its generation and evaluation process are decoupled: the model must be queried multiple times to compare candidate responses, and its evaluation requires an autoregressive reasoning process, which introduces additional computational cost. Meanwhile, [2] still requires training an additional LLM solely for evaluation.
>
> $\qquad$In contrast, RGM does not require extra forward passes nor external evaluators. The model unifies generation, reasoning, and evaluation within a single LLM and a single pass, thereby avoiding additional compute.
>
> $\qquad$[1] GenSelect: A Generative Approach to Best-of-N
>
> $\qquad$[2] Learning to Reason Across Parallel Samples for LLM Reasoning
>
> 2. Q: About comparison with Qwen-RM-72B
>
> $\quad$R: (1) Qwen-RM-72B. We select Qwen-RM-72B for comparison mainly because it serves as a representative reward model in this line of work, and our intention is to compare with representative reward models in current practice. We further use the Qwen-PRM-72B to demonstrate the effectiveness of our approach.
>
> $\qquad$(2) Additional long-reasoning baseline. For the RM specifically designed for long reasoning, we additionally train an extra ORM model using DeepScaleR-1.5B with the same training data, and the comparison results are shown as follows. Compared with it, our RGM has smaller total parameters (share backbone), simpler training pipeline (one stage training), and higher performance.
> |Method  | AIME24 | AIME25 | BRUMO25 | HMMT25 |
> | -----  |  ----  |  ----  |  ----   |  ----  |
> | ORM    |  51.7  |  33.3  |  41.7   |  18.9  |
> | RGM    |  53.1  |  35.7  |  43.2   |  21.5  |
>
> $\qquad$(3) Clarification on the “50M vs 72B” claim. We have updated the abstract to specify that the comparison refers to the total number of parameters involved in the complete TTS pipeline. For prior methods, this consists of the policy model plus a 72B reward model. In contrast, our approach only requires the policy model with an additional 50M SPRM head. And due to the shared feature, we only need to forward once in the backbone LLM. Updated wording in the paper:
>
> $\qquad$*Experiments show that our RGM, equipped with only 50M additional parameters in SPRM, outperforms policy models with 72B extra reward models.*
>
> 3. Q: The low performance on MCTS
>
> $\quad$R: (1) Performace on MCTS. In challenging tasks with long reasoning trajectories, MCTS needs to explore very deep search levels, causing the total number of nodes to grow exponentially with depth. Under a limited computation budget, the search must be stopped early, which both restricts the explored space and risks missing errors that appear in later reasoning steps, leading to degraded performance. In contrast, Best-of-N (repeated sampling) evaluates multiple full-length reasoning trajectories directly within the same budget, which often yields better results. Notably, [1] empirically demonstrates that MCTS underperforms Best-of-N under limited compute, as shown in Fig. 4 and Fig. 6 of their paper.
>
> $\qquad$(2) Process reward ability. Results in Table.4 of our paper and the ablation study in Weakness.2 have shown the SPRM with process information has better performance. This suggests that step-level reward signals bring benefits beyond outcome-based scoring.
>
> $\qquad$[1] Wider or Deeper? Scaling LLM Inference-Time Compute with Adaptive Branching Tree Search. NIPS2025

---

> > ### Author Response · Authors · 2025-11-26
> >
> > **Questions**:
> >
> > 1.  Q: About "Aha moment"
> >
> > $\quad$R: (1) "aha moment" surprising. As shown in Fig. 5, at the beginning of training, SPRM's predictions and the pseudo labels in SPRLoss are both biased, causing optimization for correct and incorrect reasoning trajectories to initially move in the same direction. This creates a risk of overfitting under noisy self-supervision. However, we observe the "aha moment" which breaks this bias: There is a step where the score for correct trajectories continues to improve while the score for incorrect trajectories begins to decline. This transition indicates that the model is no longer merely following the noisy supervision but has begun internally resolving the bias and learning to discriminate on its own.
> >
> > $\qquad$(2) Training dynamics. Moreover, all model scales exhibit this aha moment (Fig.5), indicating that it is a systematic property of the RGM framework rather than a random fluctuation in training.
> >
> > 2. Q: Comparison with ORM trained on the same data
> >
> > $\quad$R: See the response to Weakness.2
> >
> > 3. Q: Comparison with majority voting
> >
> > $\quad$R:  See the response to Weakness.1

---

### Official Review · Reviewer_sWYC · 2025-11-04

**Soundness:** 2
**Presentation:** 3
**Contribution:** 2
**Rating:** 4
**Confidence:** 4

**Summary:**

This work aims to improve the test-time performance of policy models. Specifically, the focus is on allowing the policy to generate multiple candidates for a given query and then using a process reward model to select the best candidate from the pool. The key innovation of this paper is sharing the backbone network between the policy and the process reward model to reduce parameter overhead. Additionally, the authors propose a method that leverages the consistency between final answer correctness and the scores generated by the process reward model, in order to mitigate the negative effects caused by false positive and false negative samples during the training of the process reward model.

**Strengths:**

- By sharing the backbone network, the proposed method reduces the inference cost of using the PRM to evaluate policy rollouts.

- Experimental results show that the proposed SPRM achieves superior performance with the addition of fewer parameters.

- The proposed method is simple and seems to be effective.

**Weaknesses:**

- Missing related work on process reward models. Several studies [1-4] also incorporate outcome labels to train a process reward model, which is highly relevant to this paper.

- Other work [5] has introduced '\n' as a step token. What is the rationale and benefit behind selecting '\n\n' instead? A concern is that if the policy model does not generate '\n\n', how would this method remain applicable?

- Regarding line 219, I have a concern about the clarification: "Since the representation in the last layer mainly captures the logits prediction for a single token, we use the hidden representations from the second-to-last layer of the policy model to provide richer contextual information." Why does the second-to-last layer provide richer contextual information than the last layer? More theoretical or empirical justification is needed for this assertion.

[1] From r to Q: Your Language Model is Secretly a Q-Function, COLM 2024.

[2] Discriminative Policy Optimization for Token-Level Reward Models, ICML 2025.

[3] DPO Meets PPO: Reinforced Token Optimization for RLHF, ICML 2025.

[4] Free Process Rewards without Process Labels, ICML 2025.

[5] Let's Verify Step by Step, Arxiv 2023.

**Questions:**

- What does the variable c represent in the Linear(c, 2c) at Line 206?

- In Equation 6, what do the terms $y_i$ and $Score_i$ with the index $i$ represent?

---

> ### Author Response · Authors · 2025-11-26
>
> Dear reviewer,\
> $\qquad$Thank you for your constructive comments. Below, we will respond to weaknesses and questions point by point.
>
> **Weaknesses**:
>
> 1. Q: Related works
>
> $\quad$R: Thank you for your supplement. We have added a discussion section in the paper：
>    Recent works [1-4] also generate process reward with only final answers. However, [1,3] require the reference model in Reinforcement Learning to help calculate the reward, which is mainly used for improving the training efficiency of the policy model. [2,4] require training additional LLM-based reward models (e.g., Llama-3-70B-Instruct in [2] and ImplicitPRM-8B in [4]).
>
> $\qquad$Overall, these methods still depend on external LLMs as the reward model, thus assigning the process evaluation capability to an additional model. In comparison, our method unifies the process reward model and the policy model within a single LLM, thereby integrating both reasoning and evaluation capabilities into a single model. This eliminates the need for additional reward models and streamlines the training and inference pipeline.
>
> $\qquad$[1] From r to Q: Your Language Model is Secretly a Q-Function, COLM 2024.
>
> $\qquad$[2] Discriminative Policy Optimization for Token-Level Reward Models, ICML 2025.
>
> $\qquad$[3] DPO Meets PPO: Reinforced Token Optimization for RLHF, ICML 2025.
>
> $\qquad$[4] Free Process Rewards without Process Labels, ICML 2025.
>
> 2. Q: Why use '\n\n' as the step-token
>
> $\quad$R: (1) Motivation. For LLMs generating long CoT reasoning trajectories, using \n splits the reasoning into very fine-grained steps, which may be unnecessarily detailed. In contrast, \n\n naturally serves as paragraph-level segments and appears consistently in long-chain-of-thought reasoning, better aligning with the intended definition of step segmentation. This choice is also consistent with prior work, e.g., Qwen2.5-Math-PRM, which uses \n\n as a step separator.
>
> $\qquad$(2) Responses without \n\n. For long reasoning LLMs, Responses without \n\n are typically very short answers that do not require complex reasoning. We argue that for these scenes, the LLM is unlikely to generate low-quality results, and therefore, TTS-based scoring is unnecessary.
>
> 3. Q: Evidence of using the second-to-last layer
>
> $\quad$R: (1) Theoretical. First, the last layer must align its representation with the lm_head weights to produce token logits, which limits its contextual information.  Second, using one feature for two different classifiers can create task conflicts. In contrast, the second-to-last layer preserves richer semantics before projection and is therefore more suitable for auxiliary objectives.
>
> $\qquad$(2) Experimental. We add an ablation study with analysis of using the last layer in the paper, the results is shown as in the following table. It is seen that using the second-to-last is a better choice.
> |Method                 | AIME24 | AIME25 | BRUMO25 | HMMT25 |
> | -----                 |  ----  |  ----  |  ----   |  ----  |
> | Last layer            |  48.8  |  34.6  |  41.3   |  18.8  |
> | Second-to-last layer  |  53.1  |  35.7  |  43.2   |  21.5  |
>
>
> **Questions**:
>
> 1. Q: c in the Linear(c, 2c) at Line 206
>
> $\quad$R: We have added the definition of c in the paper. c is the channel of hidden states, which is also the input channel of the SPRM.
>
> 2. Q: Error in Equation 6
>
> $\quad$R: We apologise for the issue with the subscripts in Eq. 6. We have reconstructed Eq.6 as follows. Here, y indicates whether the current reasoning path is correct. $\hat{y_n}$ denotes the pseudo-label at step n of the reasoning path, and $Score_n$ represents the SPRM prediction score at step n of the reasoning path.
>
> $L_{SPR} = \frac{1}{N}\sum_{n=1}^{N} (I(y=\hat{y_n})) * BCELoss(\text{Score}_{n},\hat{y_n}),\quad \text{where} \ \hat{y_n} = I(\text{Score}_n>0.5)$

---

### Official Review · Reviewer_jVpd · 2025-11-06

**Soundness:** 3
**Presentation:** 3
**Contribution:** 4
**Rating:** 8
**Confidence:** 3

**Summary:**

The paper introduces reflective generative models for test-time scaling, which use a shared network for policy and process reward models. This dramatically decreases the number of extra parameters to 50M. Additionally, the paper introduces a self-supervised loss (SPR Loss), which directly learns the quality of the reasoning trajectory from the outcome reward. The core idea is to have the same model both generate reasoning trajectories and score them with minimal extra parameters. The authors conducted a wide range of experiments across baseline models and demonstrated high performance (OpenAI o3-mini level), outperforming billion-scale reward models.

**Strengths:**

- Originality: The paper proposes a highly original idea of using the same backbone for policy and reward models. This idea is a novel and exciting extension of prior reward models, which are typically large and separately trained. This idea opens up a lot of exciting directions for enabling richer interactions between reasoning trajectory generation and evaluation.
- Quality: The proposed framework is clearly defined and well-motivated. The experimental evaluation is comprehensive, including multiple models and benchmarks.
- Clarity: The text and figures are clear and easy to understand.
- Significance: The paper makes a highly significant contribution to test-time scaling.

**Weaknesses:**

1. The design of the self-supervised process reward loss (SPR loss) could benefit from additional motivation and clarification. Specifically, the binary weight w_n only includes a step in the loss when the predicted step score aligns with the final outcome. Why choose a hard threshold (0.5) vs other alternatives? Could such a hard cutoff potentially discard a large fraction of training samples, particularly early in training? And could this selective inclusion behavior relate to later observations, such as the “aha” moment? Also, a minor point: y_n is the correctness of the final answer, which shouldn’t depend on n? Why use a subscript (which could be misleading)?
2. The paper’s discussion of the “aha moment” (Sec. 5.4; Fig. 5) is vague and under-analyzed. The authors highlight a green dashed line to indicate where correct and incorrect trajectory scores begin to diverge, yet provide no quantitative criterion for identifying this point. Visually, the gap between curves appears to increase gradually rather than showing a discrete transition. If there is indeed a transition, could it be simply explained by the use of a hard 0.5 threshold in the SPR loss? The authors could add more quantitative analysis on the representation or gradient through learning if this “aha moment” is indeed an important finding.
3. The paper claims that SPRM generalizes across domains, but this claim is only supported by results on LiveCodeBench. Given that mathematics and coding reasoning tasks share very similar structures, this claim of generalization currently lacks sufficient evidence. This claim could be substantially strengthened by either including evaluation on more diverse tasks or adding a discussion on what types of domains the current approach is expected to generalize well to and where its limitations might lie. For example, the segmentation choice of using ‘\n’ might not be as suitable for tasks involving natural language?
4. The explanation for why MCTS underperforms relative to Best-of-N (BoN) is speculative and unsupported by evidence.

**Questions:**

1. The final trajectory score is defined as the geometric mean of the single-step scores (Eq. 5). It might be helpful to add a sentence on the motivation/justification for this choice. I assume the geometric mean is chosen to push all step scores to be decent, since a single low score can substantially reduce the overall score. It might be interesting to add what scenarios this choice of final score does not yet capture. For example, right now, the final score treats each single step score equally and independently. In the case where the reasoning trajectory later corrects for its earlier mistakes, the earlier bad step might undesirably penalize the entire trajectory. There is an interesting question of how to incorporate the trajectory structure into the score. Have you tried any alternative form?
2. Minor type: Eq. 6 (line 245), score_i and y_i should be score_n, y_n.

---

> ### Author Response · Authors · 2025-11-26
>
> Dear reviewer,\
> $\qquad$Thank you for your constructive comments. Below, we will respond to weaknesses and questions point by point.
>
> **Weaknesses**:
>
> 1. Q: Threshold in SPR loss, influence on early training, selective inclusion behavior, error in Eq.6
>
> $\quad$R: (1) Threshold choice. We use a hard threshold of 0.5 to convert SPRM prediction scores into pseudo labels (scores > 0.5 as positive). The main purpose is to select high-confidence steps for self-supervised training, which stabilizes learning without relying on step-level human annotations. To validate that this design does not critically depend on the threshold value, we also experimented with adaptive thresholds (mean and median), and the resulting performance is comparable to the hard threshold (see table below), indicating that the SPRLoss is robust to threshold choice.
> |Method               | AIME24 | AIME25 | BRUMO25 | HMMT25 |
> | -----               |  ----  |  ----  |  ----   |  ----  |
> | Mean as threshold   |  50.0  |  35.0  |  45.0   |  21.1  |
> | Median as threshold |  51.7  |  35.0  |  42.5   |  20.0  |
> | Hard threshold      |  53.1  |  35.7  |  43.2   |  21.5  |
>
> $\qquad$(2) Early training and sample discard. At the beginning of training, most prediction scores are low, so many positive steps are initially filtered out. However, a small number of high-confidence positive steps receive relatively large weights after sample-level averaging, ensuring normal training. As SPRM is optimized, more steps become high-confidence, gradually reducing this issue.
>
> $\qquad$(3) Relation to the “aha moment.” See the discussion in Weakness.2.
>
> $\qquad$(4) We have revised Eq.6 in the paper.
>
> 2. Q: Analysis of “aha moment”
>
> $\quad$R: Thanks for your suggestion. We have revised Section 4.4. [1][2] propose that the "aha moment" enables the model to perform self-correction and self-reflection. In RGM, as we propose an SPRM head to evaluate itself, we define an "aha moment" as the step at which the SPRM starts to discriminate the correct and incorrect reasoning trajectories. In Fig.5, at the initial phase of training, both the SPRM and the pseudo label in SPRLoss is biased. However, we observe a step at which the optimization behaviors for correct and incorrect reasoning trajectories begin to diverge, indicating that the SPRM has acquired the ability to discriminate — the "aha moment". Formally, the "aha moment" is defined as the first training step at which the slope of the curve for correct trajectories becomes positive while that for incorrect trajectories becomes negative.
>
> $\qquad$Regarding the 0.5 threshold, we think the "aha moment" cannot be mainly attributed to the hard 0.5 filtering threshold in the SPR loss. The "aha moment" is defined by a shift in optimization direction. Although the hard threshold in the SPR loss can reduce overfitting on incorrect steps in correct samples, facilitates the model's ability to discriminate correct vs. incorrect reasoning trajectories, we think it is not the decisive factor for the "aha moment" as other threshold above also can lead to similar results.
>
> $\qquad$[1] Deepseek-r1: Incentivizing reasoning capability in llms via reinforcement learning
>
> $\qquad$[2] Beyond'Aha!': Toward Systematic Meta-Abilities Alignment in Large Reasoning Models
>
> 3.Q: Generalization of SPRM
>
> $\quad$R: (1) Additional empirical evaluation. In Appendix A, we report the performance on C-EVAL, a Chinese multi-domain QA dataset.
>
> $\qquad$(2) Task characteristics and expected generalization. SPRM is optimized on complex mathematical data with long reasoning trajectories. By detecting errors along reasoning paths, it excels in tasks with long, multi-step reasoning trajectories—for example, complex mathematical or coding problems. For tasks that do not require reasoning or involve only very short reasoning steps (e.g., factual QA, simple natural language queries), SPRM is less effective because there is insufficient step-level context for error detection. In such cases, the main bottleneck is the LLM’s knowledge rather than SPRM itself.
>
> $\qquad$(3) Segmentation and limitations. Our current implementation uses .\n\n as a step separator, which works well for structured reasoning traces (math and code). It also appears stable in long CoT responses, and existing works (e.g., Qwen2.5-Math-PRM) also use a similar step separator. We acknowledge that this choice may be suboptimal for tasks with unstructured natural language or tasks without clear step boundaries. Adapting the segmentation strategy could extend SPRM to other domains, but this remains a direction for future work. We have revised the limitation section.

---

> > ### Author Response · Authors · 2025-11-26
> >
> > 4. Q: Evidence of why MCTS underperforms relative to Best-of-N
> >
> > $\quad$R: We have added further analysis in the paper. In challenging tasks with long reasoning trajectories, MCTS needs to explore very deep search levels, causing the total number of nodes to grow exponentially with depth. Under a limited computation budget, the search must be stopped early, which both restricts the explored space and risks missing errors that appear in later reasoning steps, leading to degraded performance. In contrast, Best-of-N (repeated sampling) evaluates multiple full-length reasoning trajectories directly within the same budget, which often yields better results. Notably, [1] empirically demonstrates that MCTS underperforms Best-of-N under limited compute, as shown in Fig. 4 and Fig. 6 of their paper.
> >
> > $\qquad$[1] Wider or Deeper? Scaling LLM Inference-Time Compute with Adaptive Branching Tree Search. NIPS2025
> >
> > **Questions**:
> >
> > 1. Q: Motivation and limitation of geometric mean, alternative form
> >
> > $\quad$R: (1) Motivation. Prior work [1] propose to use the product of single-step scores. However, for long reasoning trajectories, the product tends to decrease with the number of steps, causing the final score to be sensitive to trajectory length. We adopt the geometric mean to mitigate this effect, ensuring that the final score reflects the quality of each step without being dominated by the trajectory length. We have added the motivation for using the geometric mean in the paper.
> >
> > $\qquad$(2) Limitations. We agree with your concern, and some critical errors could be ignored after the geometric mean with equal weight for each step. Overall, the experiment results have shown that the geometric mean is already a good choice and incorporating the trajectory structure into the scoring function remains an interesting direction for future work.
> >
> > $\qquad$(3) Alternative forms. We also conducted experiments with the arithmetic mean and the product of scores (using only the final step score was already reported in Section 5.5). The results are summarized below:
> > |Method           | AIME24 | AIME25 | BRUMO25 | HMMT25 |
> > | -----           |  ----  |  ----  |  ----   |  ----  |
> > | Product         |  44.2  |  31.1  |  40.0   |  17.9  |
> > | Arithmetic mean |  52.9  |  35.2  |  43.3   |  21.1  |
> > | Geometric mean  |  53.1  |  35.7  |  43.2   |  21.5  |
> >
> > $\qquad$[1] Let's verify step by step. ICLR2023
> >
> > 2. Q: Error in Eq.6
> >
> > $\quad$R: We have revised Eq.6 in the paper.

---

### Official Review · Reviewer_WqcA · 2025-11-06

**Soundness:** 1
**Presentation:** 1
**Contribution:** 1
**Rating:** 2
**Confidence:** 4

**Summary:**

The paper introduces a Reflective Generative Model that unifies a policy model and a process reward model through a shared backbone. It proposes a filtering mechanism for learning a process reward module (SPRM) based on outcome rewards, aiming to eliminate the need for process-level annotation

**Strengths:**

Strong empirical validation: The experiments are extensive and include multiple benchmarks and LLMs.

**Weaknesses:**

- SPRM is a filtering mechanism, not self-supervised: The model filters step-level data via a binary weight that retains only steps consistent with the outcome.
- Limited novelty: The shared backbone between policy and reward model is an engineering optimization rather than a conceptual advance in test-time scaling. Moreover, the paper does not study whether shared parameters introduce bias.
- Terminology and clarity issues: The formulation of LLMs lacks rigor  (e.g., “basic LLM”)

**Questions:**

- What is the reasoning behind using the same backbone for the reward model and policy? Could this introduce bias or reward hacking effects?
- What is the multi-agent data cleaning framework used to obtain high-quality samples for the dataset?
- Why is SPRM described as a self-supervised method, given that it relies on filtered outcome correctness?

---

> ### Author Response · Authors · 2025-11-26
>
> Dear reviewer,\
> $\qquad$Thank you for your constructive comments. Below, we will respond to weaknesses and questions point by point.
>
> **Weaknesses**:
>
> 1. Q: The self-supervised in SPRM
>
> $\quad$R: We describe SPRM as a self-supervised model because it does more than merely filter steps consistent with the final outcome. Unlike a static filtering mechanism, SPRM generates pseudo labels from its own evolving predictions at each training step. These pseudo labels are used in SPRLoss to refine coarse outcome-level supervision into step-level supervision. Importantly, as SPRM trains, its predictions—and therefore the pseudo labels—continuously update, allowing the model to iteratively improve its ability to discriminate correct and incorrect steps.
>
> $\qquad$We have adjusted Eq. 6 and the statement in our paper (as shown below) to reduce ambiguity:
>
> $L_{SPR} = \frac{1}{N}\sum_{n=1}^{N} (I(y=\hat{y_n})) * BCELoss(\text{Score}_{n},\hat{y_n}),\quad \text{where} \ \hat{y_n} = I(\text{Score}_n>0.5)$
>
> $\qquad$where $I$ is the indicator function, $n$ denotes the step-tokens, $\text{Score}_{n}$ is SPRM's process score on step $n$, $\hat{y_n}$ is the pseudo label from SPRM on step $n$, and $y$ denotes whether the final answer from the policy model is correct.
>    Because $\hat{y}_n$ evolves during training, SPRM does not simply filter data; it self-supervises its learning, gradually refining step-level evaluation. This iterative self-improvement is the key distinction that justifies calling SPRM a self-supervised module rather than a static filter.
>
> 2. Q: The novelty and whether shared parameters introduce bias
>
> $\quad$R: (1) Novelty. The main conceptual contribution of our work lies in the Reflective Generative Form (RGM), which unifies solution generation and self-evaluation within a single model. Unlike prior External TTS methods that rely on separate external reward models, RGM enables step-level self-assessment without additional models, significantly simplifying the training and inference pipeline while reducing parameter overhead. This is not merely an engineering optimization, but a conceptual advance for the External TTS.
>
> $\qquad$(2) Shared parameters. To evaluate whether sharing parameters introduces bias, we add an ablation study comparing RGM with an independent reward model. As shown in the table below, the performance of the shared-parameter RGM is comparable to the independent model across multiple benchmarks, while using only 5M extra parameters compared to 1.5B for the independent reward model. The small performance gap (≤2% absolute) indicates that the bias introduced by parameter sharing is minor and does not affect the main conclusions. Furthermore, using a single shared model greatly reduces both training and inference costs, highlighting the efficiency advantage of our approach.
> |Method             | Extra Param | AIME24 | AIME25 | BRUMO25 | HMMT25 |
> | -----             |  ----       |  ----  |  ----  |  ----   |  ----  |
> | Deepscaler        |   -         |  43.1  |  30.0  |  37.4   |  19.3  |
> | independent RGM   |  1.5B       |  54.7  |  36.7  |  43.8   |  21.9  |
> | RGM               |   5M        |  53.1  |  35.7  |  43.2   |  21.5  |
>
> 3. Q: Terminology and clarity issues
>
> $\quad$R: Thank you for the suggestion. We checked and corrected the corresponding contents. (e.g., "basic LLM" to "LLMs without TTS")

---

> > ### Author Response · Authors · 2025-11-26
> >
> > **Questions**:
> >
> > 1. Q: Reasoning behind our method, bias, and reward hacking.
> >
> > $\quad$R: (1) Reasoning behind shared backbone. Sharing the same backbone for the policy and reward model significantly reduces additional parameters and computation, enabling a lightweight and efficient test-time scaling framework. Prior works such as LLM-as-a-judge and Self-Refine[1] demonstrate that strong LLMs can simultaneously generate outputs and assess their quality, suggesting that a single model has the potential to learn both generation and evaluation. The ablation study in Weakness.2 further confirms that sharing parameters achieves comparable performance to using an independent reward model, indicating that the shared backbone is feasible without compromising effectiveness.
> >
> > $\qquad$(2) Bias. See the response to Weaknesses.2
> >
> > $\qquad$(3) Reward hacking. During training, the generation policy is optimized solely based on final answer correctness. The SPRM scores are not used as a reward signal in GRPO, ensuring that the policy cannot over-optimize for SPRM outputs. This design guarantees that reward hacking is effectively prevented.
> >
> > $\qquad$[1] Self-refine: Iterative refinement with self-feedback. NIPS2023
> >
> > 2. Q: The multi-agent data cleaning framework
> >
> > $\quad$R: We used two agents for data filtering sequentially. The first agent is a difficulty classification model trained on the MATH dataset, which is used to filter out easy data. For the second agent, we use the pass rate to sample valuable data, following [1]. All data are sampled from open-source datasets, and we do not modify their content.
> >
> > $\qquad$We will release all training data to support reproducibility and enable further research by the community.
> >
> > $\qquad$[1] Exploring the Limit of Outcome Reward for Learning Mathematical Reasoning
> >
> > 3. Q: The self-supervised in SPRM
> >
> > $\quad$R: See the response to Weaknesses.1

---

### Author Response · Authors · 2025-11-26

Dear Reviewers and AC,\
$\qquad$We thank all reviewers for their constructive comments. We carefully concern each comment and reply to the concerns of each reviewer point-by-point. We understand the significance of offering additional supplements and clarifications to ensure the validity of our paper. We revise our paper based on the suggestions from all reviewers, as we firmly believe that these revisions will improve the overall quality of our work and contribute to future works.

---

### Author Response · Authors · 2025-12-03
**The concise summary of our rebuttal**

Dear AC,

Thank you for handling our submission. We have provided point-by-point responses to all the weaknesses and questions raised by each reviewer in our comments. Below is a concise summary of our rebuttal, highlighting the key clarifications and the reviewers’ consensus.

### **1. Regarding the Reject Reviewer (WqcA)**
**Main concerns:** novelty and the name of one sub-module is not suitable.
- **Novelty:** All other five reviewers **did not** raise any weaknesses or questions about limited novelty. Besides, Reviewer jVpd stated that *“This idea is a novel and exciting extension of prior reward models”*
- **Name:** we have clarified the reason in our comments. And we think this does not affect the technical correctness or contribution of our work.

### **2. Regarding Borderline Reject Reviewers (sWYC, RgpY)**
**Main concerns:** additional related works and ablation studies.
- We have **added discussions on all requested related works and ablation experiments** in the rebuttal (see our comments and revised paper).
- They also acknowledged the **effectiveness** of our method:
  - sWYC: *“Experimental results show that the proposed SPRM achieves **superior performance with the addition of fewer parameters**.”*
  - RgpY: *“**Good** empirical results ... The experimental setup is **thorough** ...”*

### **3. Regarding Accept & Borderline-Accept Reviewers (jVpd, 65ZK, uFte)**

These reviewers support our contribution:
- jVpd: *“The paper makes a **highly significant contribution** to test-time scaling.”*
- 65ZK: *“Achieving comparable performance to 72B parameter reward models with only 50M parameters is a **solid contribution**”*
- uFte: *“Clear performance improvement… **proving the method’s effectiveness**.”*

We hope this summary helps your review process. Thank you again for your time and attention to our submission.

---

### Meta-Review · Area_Chair_3VLe · 2026-01-06

**Summary:**

The Reflective Generative Model (RGM) unifies policy and reward models into a shared backbone, adding only 50M parameter. This efficiency allows a 32B model to outperform OpenAI’s o3-mini on the AIME24 benchmark. Reviewers initially have three main concerns.
1. **Novelty**. The Self-supervised Process Reward Model (SPRM) was a simple static filter rather than a truly "self-supervised" mechanism.
2. **Shared-Parameter Bias**. Concerns arose that sharing a backbone between the generator and evaluator would lead to "circular reasoning" or reward hacking.
3. **Generalization**. The scalability of using heuristic step-delimiters across different languages.
The paper has a mixed scores (2, 4, 4, 6, 6, 8). However, there is no response from reviewers during the discussion period.

**Reviewer Concerns:**

Rebuttal provided additional experiments and justifications for the concerns. The novelty is unclear.

**Reviewer Scores:**

Unfortunately, there is no discussion between authors and reviewers. If reviewers had been able to participate fully in the discussion, some may raise the score to borderline accept (e.g., from 4 to 6).

---

### Decision · Program_Chairs · 2026-01-26

Accept (Poster)